

# Genome-wide analysis reveals four key transcription factors associated with cadmium stress in creeping bentgrass (*Agrostis stolonifera* L.)

Jianbo Yuan[1,2,*], Yuqing Bai[3,*], Yuehui Chao[2], Xinbo Sun[4], Chunyan He[2], Xiaohong Liang[2], Lijuan Xie[1] and Liebao Han[2]

[1] School of Applied Chemistry and Biotechnology, Shenzhen Polytechnic, Shenzhen, China
[2] Turfgrass Research Institute, College of Forestry, Beijing Forestry University, Beijing, China
[3] Administrative Office, Wutong Mountain National Park, Shenzhen, China
[4] Key laboratory of crop growth regulation of Hebei Province, Hebei Agricultrual University, China
[*] These authors contributed equally to this work.

## ABSTRACT

Cadmium (Cd) toxicity seriously affects the growth and development of plants, so studies on uptake, translocation, and accumulation of Cd in plants are crucial for phytoremediation. However, the molecular mechanism of the plant response to Cd stress remains poorly understood. The main objective of this study was to reveal differentially expressed genes (DEGs) under lower (BT2_5) and higher (BT43) Cd concentration treatments in creeping bentgrass. A total of 463,184 unigenes were obtained from creeping bentgrass leaves using RNA sequencing technology. Observation of leaf tissue morphology showed that the higher Cd concentration damages leaf tissues. Four key transcription factor (TF) families, WRKY, bZIP, ERF, and MYB, are associated with Cd stress in creeping bentgrass. Our findings revealed that these four TFs play crucial roles during the creeping bentgrass response to Cd stress. This study is mainly focused on the molecular characteristics of DEGs under Cd stress using transcriptomic analysis in creeping bentgrass. These results provide novel insight into the regulatory mechanisms of respond to Cd stress and enrich information for phytoremediation.

## INTRODUCTION

Cadmium pollution is an environmental issue which is getting more and more attention. As a non-essential, highly toxic and easily accumulated element in plants, Cd accumulation directly inhibit plant development by negative effects on physiological and metabolic processes (*Nawrot et al., 2006*; *Hasan et al., 2009*; *DalCorso, Farinati & Furini, 2010*). Soil contamination with Cd has long been a major ecological concern worldwide in areas experiencing industrialization and urbanization (*Chen et al., 2011*; *Yu et al., 2017*).

There are two main strategies for fighting Cd pollution, phytoextraction and phytostabilization (*Salt, Smith & Raskin, 1998*). Phytoextraction involves applying a cover of vegetation at the contaminated site, thereby preventing wind and water

Corresponding authors
Lijuan Xie, xlj@szpt.edu.cn
Liebao Han, hanliebao@163.com

erosion while also developing an extensive root system that will fix soil. This method requires plants that possess tolerance to the contaminant metals and strong soil-fixing abilities. Phytostabilization employs species from the plant communities found on local contaminated sites (*Krämer, 2005*). In addition, breeding of cultivars that rapidly accumulate Cd is useful for reducing Cd pollution. For these purposes, understanding the molecular mechanisms of plant responses to Cd stress is critically important.

Creeping bentgrass (*Agrostis stolonifera* L.) is a high-value specialty grass widely used in golf courses in temperate climates around the world (*Vargas, 1993*). This turfgrass can generate a compact lawn, and this characteristic provides a powerful benefit for phytoremediation. Several previous studies have suggested that certain fast-growing, high-biomass, and metal-tolerant plant species may be effective for phytoremediation (*Juwarkar et al., 2008*). Among the molecular mechanisms of plants related to Cd stress, *Satoh-Nagasawa et al. (2011)* found that *OsHMA2* is the major transporter of Cd from roots to shoots in rice. *OsNRAMP1* participates in Cd uptake and transport at the cellular level in rice, and *OsNRAMP1* overexpression in roots may promote Cd accumulation in the shoots (*Takahashi et al., 2011*). *OsHMA3* is critical for Cd stress in rice, and mutation of this protein results in loss of Cd function into vacuoles in root cells, leading to high translocation of Cd from roots to shoots (*Miyadate et al., 2011*; *Ueno et al., 2010*; *Takahashi et al., 2011*). *OsNRAMP5* encodes a natural resistance-associated macrophage protein in rice, and functional analysis showed that a defect in this protein decreases Cd uptake by roots through the use of a mutant method in *Arabidopsis thaliana* (*Ishikawa et al., 2012*). Moreover, several transcription factors (TFs) in several families have been identified in the Cd stress response, including WRKY, ERF, MYB, and bZIP (*Wei et al., 2008*; *Tang, Charles & Newton, 2005*; *Van de Mortel et al., 2008*; *Jakoby et al., 2002*). Previous research suggests that several genes and proteins regulate uptake, translocation, and accumulation of Cd in plants.

Along with the development of '-omic' technologies in general, RNA sequencing (RNA-Seq) technology has provided a powerful approach for transcriptomic and non-coding RNA research (*Ekblom & Galindo, 2011*). *Zhang et al. (2009)* have suggested that RNA-Seq can be applied to multiple total or fractionated RNA molecules, which are converted to a library of complementary DNA (cDNA) fragments and all molecules are sequenced with a high-throughput method to obtain short sequence reads. Recently, RNA-Seq has been widely applied to analysis of differential gene expression in plant stress responses, for example, Cd stress in radish (*Xu et al., 2015*). The purpose of this study was to identify differentially expressed genes (DEGs) in creeping bentgrass under Cd stress conditions for further analysis.

Creeping bentgrass is widely used on golf courses and for urban landscaping. Previous research on this plant has mainly focused on the physiological level, such as the effects of chelated iron, biostimulants, and nitrogen on the growth of creeping bentgrass (*Ervin et al., 2004*; *Snyder, 1972*). However, the molecular mechanism of the bentgrass response to Cd stress remains unclear. In this study, we isolated a total of 49.6, 52.2, and 50.9 million clean reads from CK, BT2_5, and BT43 creeping bentgrass leaf cDNA libraries, respectively,

using RNA-Seq methods. Our study investigated the molecular characteristics of Cd stress response in creeping bentgrass.

## MATERIALS AND METHODS

### Plant material preparation

The cultivar of creeping bentgrass used in this study was A4, obtained from the forestry college of Beijing Forestry University. The seeds were soaked in water for three days. Then, seeds were sown into soil consisting of peat, vermiculite, and perlite at a ratio of 1:1:1, and containing no foreign Cd. The seedlings were grown in a greenhouse at 26 °C and plant materials used in this study were mature plants of three-month old irrigated with Hoagland nutrient solution (*Hoagland & Arnon, 1950*). For Cd stress treatments, plant materials were irrigated with nutrient solution every two days in which 2.5 mM (BT2_5) or 43 mM (BT43) $CdCl_2$ was added. Plant samples (aboveground) were harvested under treatment for 7 days. Samples were frozen in liquid nitrogen and stored at −80 °C for further experiments. All samples were taken as three biological replicates.

### Leaf tissue morphology observation

We selected CK, BT2_5, and BT43 leaves for cutting. Fresh plant tissue was placed in 4% paraformaldehyde solution for 36 h, followed by 4 h of dehydration using 75% ethanol, and 4 h of dehydration using 90% and 95% ethanol. Then the plant material was converted into a paraffin section using a microtome. The thickness of the slice was 4 μm, and it was stained with saffron and fast green FCF. Finally, the slice was used for light microscope observation (Nikon ECLIPSE-ci).

### Plant RNA isolation and cDNA preparation

Three cDNA libraries were constructed for RNA sequencing. Total RNA of CK, BT2_5, and BT43 were extracted using extraction kits (TIANGEN, Beijing, China). The First Strand cDNA Synthesis Kit as well as the PrimeScript[TM] RT Reagent Kit with gDNA Eraser (Takara, Beijing, China) were used. The reverse transcriptional reaction used 1 μl of total RNA.

### Sequencing, data filtering, and transcript assembly

The CK, BT2_5, and BT43 cDNA libraries were used for high-throughput sequencing (Illumina HiSeq[TM]), and raw reads were obtained. After verifying the quality of all raw reads, the $Q_{phred}$ value was determined, representing base quality in four grades at 90% (Q10), 99% (Q20), 99.9% (Q30), and 99.99% (Q40). Raw reads were filtered to remove adapters, uncertain base identifications, and $Q_{phred}$; no more than 20 reads were removed. We obtained clean reads by filtering, which were then used for assembly using Trinity (*Grabherr et al., 2011*). After the transcript sequence was obtained, transcript sequences were arranged into clusters by comparison using Corset (https://code.google.com/p/corsetproject/). Finally, transcript and cluster sequence lengths were used for analysis.

### Annotation of DEGs

The clean reads were mapped in RSEM software (*Li & Dewey, 2011*) to obtain read counts. From the read counts, the density distribution was determined using fragments per kilobase

of transcript per million mapped reads (*Trapnell et al., 2010*), with an *E*-value threshold of greater than 0.3. DEGs were identified using DESeq analysis (*Anders & Huber, 2010*). The padj threshold was no more than 0.05. Then, volcano plots and Venn diagrams were used for DEG analysis, and the K-means and SOM methods were used for clustering DEGs.

### Gene functional analysis and classification

To fully understand the functions of unigenes, a bioinformatics approach was used to annotate the unigenes. First, sequences were aligned based on four data libraries, including the NCBI non-redundant protein sequences (Nr), NCBI nucleotide sequences (Nt), Protein family (Pfam), and Swiss-Prot databases. The *E*-value thresholds of Nr (diamond v0.8.22), Nt (NCBI blast 2.2.28+), and Swiss-Prot (diamond v0.8.22) were 1e-5, and the E-value threshold of Pfam (HMMER 3.0) was 0.01. Second, based on the results of the Nr and Pfam annotation, Gene Ontology (GO) was used to analyze unigene functions (*Götz et al., 2008*), and the *E*-value threshold for GO (Blast2GO v2.5) was 1e-6. Finally, the Kyoto Encyclopedia of Genes and Genomes (KEGG) and eukaryotic ortholog groups (KOG) were used to further annotate unigene functions. The *E*-value threshold used for KEGG (KEGG Automatic Annotation Server) was 1e–10 and KOG (diamond v0.8.22) was 1e–3.

### Real-time quantitative polymerase chain reaction (qRT-PCR) validation

Total RNA from CK, BT2_5, and BT43 were extracted using the method described above following qRT-PCR analysis. The method used for first strand cDNA synthesis was the same as described above, with primers designed using Primer Premier 5.0 (Premier Biosoft International, Palo Alto, CA, USA), shown in Table S13. 18S RNA was used as an internal control. The BioRadC1000Server system was used to perform qRT-PCR. The reaction procedure was: 40 cycles of 95 °C for 10 min, 95 °C for 15 s, and 60 °C for 1 min, and then melting curve data was obtained. All reactions were run with technical duplicates. The $2^{-\Delta\Delta Ct}$ method was used to calculate relative expression levels (*Livak & Schmittgen, 2001*).

## RESULTS

### Leaf tissue morphological analysis under Cd stress

To elucidate the effect of Cd treatment at various concentrations on leaf cellular, histological sections were observed using a light microscope, as shown in Fig. 1. The histological results revealed that the morphological characteristics of the leaves changed markedly under the CK, BT2_5, and BT43 conditions. CK cells were morphologically normal, complete, and healthy, whereas the morphology of cells treated with BT2_5 and BT43 was anomalous. Both BT2_5 and BT43 cells were damaged, and BT43 exhibited a greater degree of damage than BT2_5. These histological observations suggested that Cd toxicity affected the morphological of plant tissues, and that higher concentrations of Cd caused more serious  effects.

### RNA sequencing and transcriptome assembly

CK, BT2_5, and BT43 leaves were selected for RNA sequencing. All data were generated using three biological replicates. RNA sequencing data quality is shown in Table 1. More

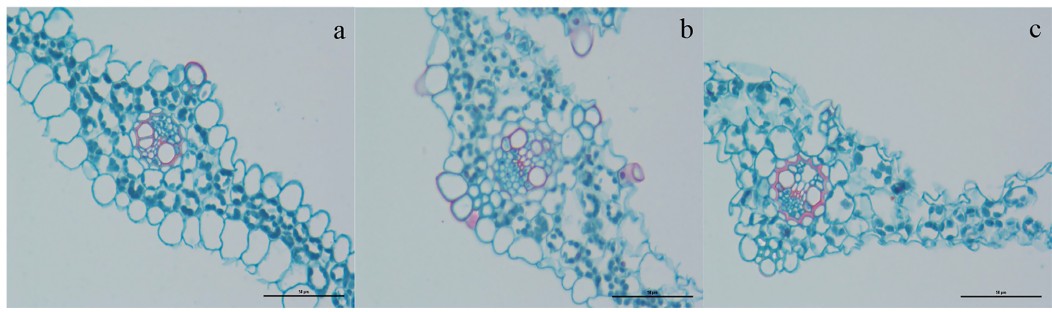

**Figure 1** **Leaf tissue morphology.** (A) CK; (B) BT_2.5; (C) BT43, bar = 50 μm, the leaf tissue stained with saffron and fast green FCF, CK cells represent morphologically, BT_2.5 cells represent minor damage, BT_43 cells represent severe damage.

**Table 1** **RNA sequencing data quality profile.**

| Sample | Raw reads | Clean reads | Clean bases | Error (%) | Q20 (%) | Q30 (%) | GC (%) |
|---|---|---|---|---|---|---|---|
| CK | 51,167,908 | 49,565,106 | 7.43G | 0.02 | 95.91 | 89.99 | 57.82 |
| BT_2.5 | 55,312,134 | 53,203,108 | 7.98G | 0.01 | 97.63 | 93.75 | 57.71 |
| BT43 | 52,608,602 | 50,981,914 | 7.65G | 0.02 | 96.07 | 90.35 | 58.34 |

**Notes.**

Q20, bases with Phred scores greater than 20 as a percentage of total bases; Q30, bases with Phred scores greater than 30 as a percentage of total bases; GC, G + C content.

than 50 million raw reads were generated for each sample, and after filtering more than 49 million clean reads remained. The false discovery rate of all data is less than or equal to 0.02. All clean raw data were assembled. The transcript and unigene length distribution is shown in Fig. 2. The mean transcript length was 473 bases, and the mean unigene length was 466 bases. Transcript N50 was 533 bases and unigene N50 was 466 bases (Figs. 2A and 2B).

## Gene functional annotation and classification

All-unigenes were searched for annotations against various databases. From a total of 463,184 sequences retrieved from different databases listed in Table 2, 203,561 were annotated in NR (43.94%). The mapping rate of NCBI nucleotide sequences (NT) was 167,084 (36.7%). A total of 111,508 sequences were annotated in Swiss-Prot (24.07%) and 27,394 transcripts were mapped in the KOG databases into 26 categories, shown in Table S1 . Most sequences clustered into 'Posttranslational modification, protein turnover, chaperones', which accounted for 14.41% (4,431), followed by 'General function prediction only' (3,402, 11.06%), 'Translation, ribosomal structure and biogenesis' (3037, 9.87%), and 'Intracellular trafficking, secretion, and vesicular transport' (2,166, 7.04%). The 'Unnamed protein' cluster was the least represented group, with only one gene included.

GO assignments were used to classify the gene functions of assembled transcripts from creeping bentgrass using Blast2GO. In all, 556,359 transcripts were mapped, and the GO databases provided 56 terms at the second level (Fig. S1). The 'cellular process'
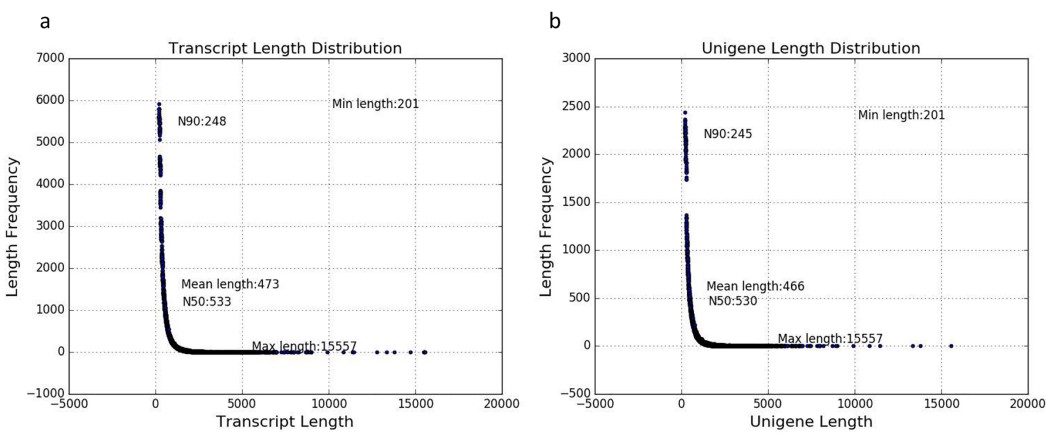

**Figure 2 Sequence distribution of transcripts and unigenes.** (A) transcript length; (B) unigene length. N50: splicing transcripts in descending order of length, accumulating transcript lengths, to a length of not less than 50% of the total spliced transcripts; N90: 90% of the total length of the spliced transcript length.

**Table 2 Gene annotation statistics.**

|  | Number of unigenes | Percentage (%) |
| --- | --- | --- |
| Annotated in NR | 203,561 | 43.94 |
| Annotated in NT | 167,084 | 36.07 |
| Annotated in KO | 59,855 | 12.92 |
| Annotated in SwissProt | 111,508 | 24.07 |
| Annotated in PFAM | 123,967 | 26.76 |
| Annotated in GO | 127,047 | 27.42 |
| Annotated in KOG | 27,394 | 5.91 |
| Annotated in all Databases | 14,496 | 3.12 |
| Annotated in at least one Database | 260,356 | 56.21 |
| Total Unigenes | 463,184 | 100 |

term represented the largest cluster in the biological process category, accounting for 11.09% (61,687), followed by 'metabolic process' (60,277, 10.83%) and 'single-organism process' (43,345, 7.79%). In the cellular component, 'cell' (31,410, 5.65%), 'cell part' (31,387, 5.64%), and 'macromolecular complex' (18,918, 3.4%) were the most abundant categories. The 'binding' term was most represented within the molecular function category, accounting for 12.57% (69,935), followed by 'catalytic activity' (51,940, 9.34%) and 'transporter activity' (5,904, 1.06%). Moreover, 'transcription factor activity, protein binding', 'transporter activity', 'response to stimulus', and 'negative regulation of biological process' were also identified in creeping bentgrass under Cd stress. Detailed GO data are shown in Tables S2 and S3.

KEGG was also used to identify the gene biological pathways. The assembled transcripts were searched against the KEGG database (http://www.genome.jp/kegg/) with an *E*-value cutoff of 1e–10. Overall, 48,400 transcripts were involved in 130 KEGG pathways

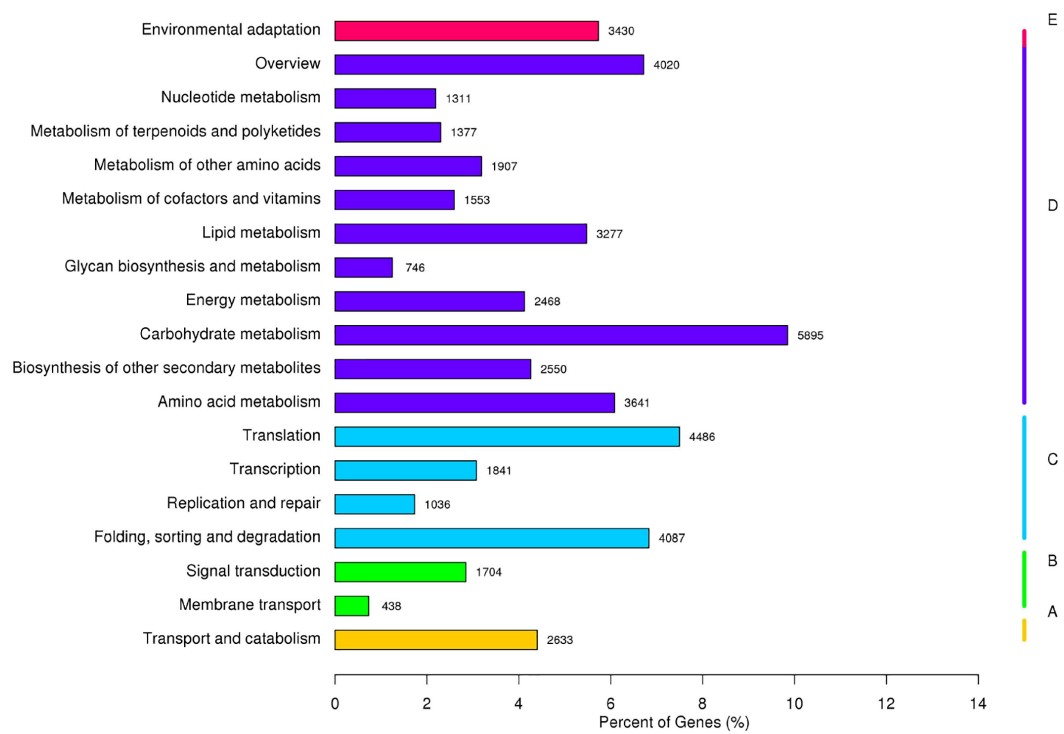

**Figure 3   KEGG classifications.** (A–E) for Cellular Processes, Environmental Information Processing, Genetic Information Processing, Metabolism, and Organismal Systems, respectively.

(Table S3). All KEGG terms at the second level are shown in Fig. 3. 'Carbohydrate metabolism' was the highest category, accounting for 12.18% (5,895). In addition, several categories exhibited a high ratio, such as 'Environmental adaptation' (3,430, 7.09%), 'Folding, sorting and degradation' (4,087, 8.44%), and 'Transport and catabolism' (2,633, 5.44%). These categories may play important roles in response to Cd stress.

## Functional annotation of DEGs involved in Cd stress

In the present study, clean reads from the CK, BT2_5, and BT43 libraries were mapped to transcriptome reference sequences using DESeq (*Anders & Huber, 2010*). Unigenes that met the screening thresholds of $|log_2FC| >1$ and $p < 0.005$ were confirmed as DEGs. Comparison of the CK and BT2_5 libraries indicated that 90 DEGs were upregulated and 189 DEGs were downregulated (Fig. 4A), while 1,175 DEGs were upregulated and 1,334 DEGs were downregulated between the BT2_5 and BT43 libraries (Fig. 4B). Moreover, comparing CK with BT43 showed that 1730 DEGs were upregulated and 1761 DEGs were downregulated (Fig. 4C).

GO enrichment analysis was performed to obtain functional annotations of the DEGs. Comparison of CK with the BT2_5 library exhibited that 'catalytic activity (GO:0003824)' was the predominant GO term in the molecular function category, including 10 upregulated genes. The dominant GO term in the biological process group was 'metabolic process (GO:0008152)', which included 14 upregulated genes. The 'membrane (GO:0016020)'

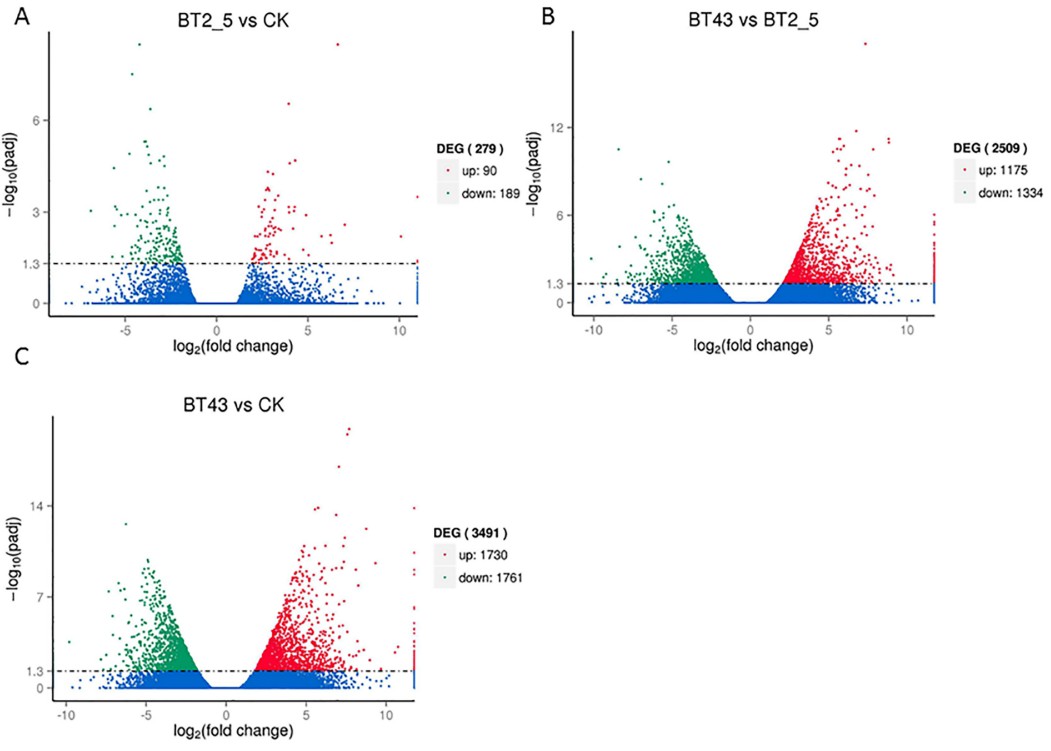

**Figure 4  Volcano plots comparing differentially expressed genes (DEGs).** (A) DEGs between BT_2.5 and CK; (B) DEGs between BT43 and BT_2.5; (C) DEGs between BT43 and CK. Red dots represent up-regulated genes; Green dots represent down-regulated genes; blue dots represent genes that have not been changed.

GO term was predominant among cellular components, including seven upregulated genes (Table S3). In the BT2_5 and BT43 library comparison, the predominant GO terms included 'metabolic process (GO:0008152, 414)' in biological processes, 'catalytic activity (GO:0003824, 385)' in molecular functions, and 'membrane (GO:0016020, 135)' in cellular components (Table S3). Comparing BT43 with CK showed that the predominant GO terms were 'metabolic process (GO:0008152, 592)' in biological processes, 'catalytic activity (GO:0003824, 556)' in molecular functions, and 'cell (GO:0005623, 186)' in cellular components (Table S3). Among the upregulated GO terms, the predominant GO terms between CK and BT2_5 were 'metabolic process (GO:0008152, 59)' in the biological process group, 'binding (GO:0005488, 55)' in the molecular function group, and 'cell (GO:0005623, 30)' in the cellular component category (Table S3). Comparing the BT43 and BT2_5 libraries indicated that the GO terms 'metabolic process (GO:008152, 419)' in biological processes, 'catalytic activity (GO:0003824, 389)' in molecular functions, and 'membrane (GO:0016020, 113)' in cellular components were predominant (Table S3). The 'metabolic process (GO:0008152, 634)' term in biological processes, 'catalytic activity (GO:0003824, 575)' in molecular functions, and 'cell (GO:0005623, 278)' in cellular components dominated the comparison between CK and BT43 (Table S3). In addition,

the 'metal ion binding', 'transport', and 'UDP-glycosyltransferase activity' terms were also active during Cd stress in creeping bentgrass.

To further determine how these DEGs are involved in biological metabolic pathways, KEGG analysis was implemented to identify pathways associated with Cd stress using the hypergeometric test method. In all, 152 pathways were upregulated and 182 pathways were downregulated among various libraries. Comparing CK and BT2_5 showed that the 'Galactose metabolism (ko00052)' and 'Starch and sucrose metabolism (ko00500)' terms were predominant among upregulated pathways (Table S4), while the 'Plant-pathogen interaction (ko04626)', 'Cutin, suberin and wax biosynthesis (ko00073)', and 'Peroxisome (ko04146)' terms accounted for most genes in downregulated pathways (Table S5). In the BT2_5 and BT43 libraries (Table S4 , Table S5), the predominant upregulated pathways were 'Glutathione metabolism (ko00480)', 'Plant hormone signal transduction (ko04075)', and 'Phenylpropanoid biosynthesis (ko00940)', and the predominant downregulated pathways were 'DNA replication (ko03030)', 'Phenylpropanoid biosynthesis (ko00940)', and 'Isoquinoline alkaloid biosynthesis (ko00950). Comparison of BT43 with CK showed that 'Starch and sucrose metabolism (ko00500)', 'Galactose metabolism (ko00052)', and 'Plant hormone signal transduction (ko04075)' hold predominant positions in upregulated pathways (Tables S4 and S5), whereas the predominant positions in downregulated pathways were 'Photosynthesis - antenna proteins (ko00196)', 'Starch and sucrose metabolism (ko00500)', and 'Galactose metabolism (ko00052)'. In addition, several pathways were also activated in creeping bentgrass under Cd stress, such as 'Glutathione metabolism (ko00480)', 'RNA degradation (ko03018)', and 'ABC transporters (ko02010)'. These pathways were important for creeping bentgrass resistance to Cd stress.

## Several functional genes are involved in signal transduction pathways

Mitogen-activated protein kinase (MAPK) pathways represent a signaling mechanism in the plant response to Cd stress that includes MAPK, MAPK kinase (MAPKK), and MAPK kinase kinase (MAPKKK). The MAPK cascade plays critical roles in the plant response to Cd stress. Under Cd stress, many kinases must be activated, and these kinase enzymes belong to the MAPK family. In the present study, MAPKKK3 and MAPKKK12 were involved in the bentgrass response to Cd stress. In the CK to BT43 comparison, MAPKKK3 was downregulated (Table S6). On the other hand, MAPKKK12 was upregulated from CK to BT43.

Glutathione S-transferase (GST) is important for plant adaptation to a variety of stressful conditions, because GST promotes the scavenging of reactive oxygen species (ROS). In our study, several GSTs were obtained, including GSTU17, GSTU6, GSTU22, GST3, GST4, and GST23. GSTU17 was downregulated from CK to BT2_5, whereas it was upregulated from BT2_5 to BT43 (Table S7). The other 'GSTU-' enzymes were consistent with GSTU17. In addition, GST4 and GST23 were upregulated in all RNA-Seq libraries. Salicylic acid (SA) is important in a well-known hormonal signal transduction pathway related to metal stress. SA can provide plants with the protective ability to resist unfavorable conditions. Our results indicated that salicylic acid-binding protein (SABP)-2 was active, as SABP2 was

upregulated in all libraries (Table S8). In summary, some signal transduction pathways play critical roles in the creeping bentgrass response to Cd stress.

## WRKY TFs involved in Cd stress

The WRKY gene family in plants contains specific transcriptional regulators, which play important roles in plant responses to stress. WRKY proteins bind W-boxes to regulate many stress-related genes (*Eulgem et al., 1999*). In our study, several WRKY TFs exhibited significant differential expression under the different Cd treatment conditions, including *WRKY33*, *WRKY12*, *WRKY23*, *WRKY75*, *WRKY2*, and *WRKY53* (Table S9). In the BT2_5 treatment, *WRKY33* was downregulated, but was upregulated following BT43 treatment. *WRKY12* and *WRKY2* were also upregulated following BT2_5 treatment, but these TFs were downregulated in the BT43 treatment concentration. *WRKY23* and *WRKY75* were upregulated under all treatment conditions. On the other hand, *WRKY53* was downregulated in all treatments. KEGG pathway enrichment exhibited that *WRKY33* and *WRKY2* were involved in 'Plant-pathogen interaction' pathway (ko04626). The role of *WRKY33* is to suppress the induction of defense genes. The role of *WRKY2* is to promote defense related-gene and accelerate programmed cell death.

## Response of basic leucine zipper motif (bZIP) TFs to Cd stress in bentgrass

The leucine zipper dimerization motif (Box 1) is a basic region found in the bZIP family. The bZIP TFs play important roles in stress signal transduction. RNA-Seq results indicated that three DEGs belonging to the bZIP family are involved in Cd stress in bentgrass, including basic leucine zipper 6 (*BZIP06*), *BZIP43*, and *BZIP19*. All bZIP TFs were upregulated in the BT2_5 treatment, but *BZIP06* and *BZIP19* were downregulated in the BT43 treatment (Table S10). Only *BZIP43* maintained upregulation in BT43.

## Ethylene-responsive factor (ERF) TFs related to Cd stress

The ethylene-responsive factor (ERF) subfamily belongs to the APETALA 2 (AP2) family, which responds to abiotic stresses in plants, including ethylene, drought, and high salinity. In our study, several ERF genes were observed under Cd stress in bentgrass, including *ERF1B*, *ERF115*, *ERF110*, *ERF7*, *ERF113*, *ERF4*, and *ERF15*. In the BT2_5 Cd treatment, most genes were upregulated, while *ERF115* and *ERF4* were slightly downregulated. In the BT43 treatment, most genes were upregulated, including *ERF1B*, *ERF115*, and *ERF15*. The $\log_2$ ratios of these genes exhibited more than five-fold changes. Only *ERF4* was slightly downregulated in BT43 (Table S11). KEGG pathway enrichment showed that *ERF1B* was involved in 'Ethylene' pathway (ko04075) and *ERF1B* plays an important regulatory role in promoting senescence.

## MYB TF responses to Cd stress

MYB TFs are key factors in the regulation of development, metabolism, and responses to abiotic stresses in plants. The RNA-Seq results indicated that four DEGs in the MYB family were involved in the Cd stress response in bentgrass, including *MYB4*, *MYB39*, *MYB108*,

and *MYB305*. Most of these genes were upregulated in the BT2_5 and BT43 libraries (Table S12). Only *MYB4* was downregulated in BT_25, but this gene was upregulated in BT43.

## Validation of DEGs via qRT-PCR

To confirm the differential expression patterns of genes identified in the RNA-Seq data, nine candidate genes were randomly selected for qRT-PCR. The expression level of each gene from CK, BT2_5, and BT23 was compared with its abundance based on the RNA-Seq data. The results indicated that the expression levels of all genes were downregulated in the BT2_5 Cd treatment, including *CPIP9*, *NFYB5*, and *RCCR* (Fig. 5). The expression levels of these genes were downregulated more than three-fold. However, most candidate genes were upregulated in the BT43 Cd treatment, including *NFYB5*, *RCCR*, and *HOX22*. Moreover, several genes exhibited downregulation in the BT43 treatment, including *GOS9*, *CYP71Z6*, and *ATP*. In summary, the trend changes of most genes determined by qRT-PCR were consistent with the RNA-Seq results. A few genes exhibited different trends in qRT-PCR and RNA-Seq data, which may be attributed to the sensitivity of qRT-PCR.

## DISCUSSION

### Higher Cd concentrations damage leaf tissue

Toxic heavy metal contamination in plants has attracted major concern in recent years because plants, including crops, can affect animal and human health (*Nan et al., 2002*). Cd is an unessential element for plants. However, Cd can interact with other elements, such as zinc (Zn), copper (Cu), and manganese (Mn), and affect their uptake and translocation (*Lachman et al., 2015*). Some studies have reported that Cd can affect plant development in various plant species by inhibiting the absorption of water and nutrients, resulting in various symptoms of injury *in vivo* or *in vitro* (*Li et al., 2008*). Cd can interfere with several photosynthetic complexes, resulting in reduced photosynthetic carbon assimilation (*Maksymiec, Wojcik & Krupa, 2007*). Moreover, Cd affects guard cell regulation via calcium channels, thus disrupting the plant's water status (*Perfus-Barbeoch et al., 2002*). Higher Cd concentrations have been shown to reduce seed germination and root and shoot elongation in *Medicago sativa* L. (*Peralta et al., 2001*). *Wang et al. (2007)* revealed that Cd interacts with other metals (Mn, Fe, Cu, and Zn) that may accumulate in the roots and shoots of *Zea mays* L. Cd caused significant inhibition of growth in *Pisum sativum* L. roots and leaves by reducing the rate of photosynthesis and the chlorophyll content of leaves, and altering the nutrient status (*Sandalio et al., 2001*). Cd induced production of ROS, $H_2O_2$, and $O_2^-$ in pea leaves, and these signal molecules activate several defense genes against Cd toxicity (*Romero-Puertas et al., 2004*). In our study, 2.5 mM (BT2_5) and 43 mM (BT43) Cd damaged leaf tissue in bentgrass (Figs. 1B and 1C), while plants given 0 mM (CK) remained healthy after seven days. RNA-Seq data showed that higher Cd concentrations affected the water uptake and nutritional status of leaves, causing a tissue morphological disorder in bentgrass. These results were consistent with previous reports.
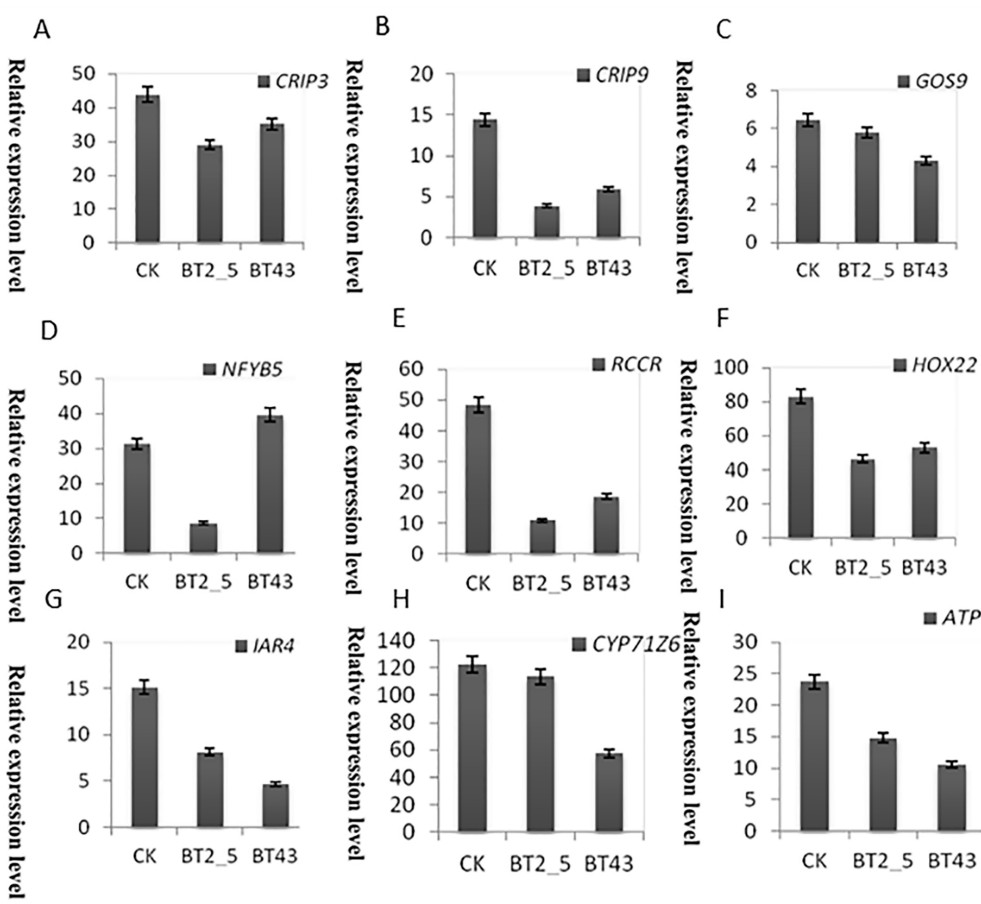

**Figure 5** **Validation of the selected DEGs using qRT-PCR under CK, BT_2.5, and BT43 Cd treatment conditions.** The relative expression of each gene was determined from three technical replicates using the $2^{-\Delta\Delta Ct}$ method.

## MAPK cascades play active roles in the response to Cd stress

The plant response to Cd stress involves changes in the expression patterns of many genes. To adapt to difficult environmental conditions, specific genes were activated by plant cells. Signal transduction pathways can drive differential gene regulation (*DalCorso, Farinati & Furini, 2010*).

MAPK cascades occur in response to various osmotic stresses, causing hyperosmotic changes in the volume and turgor pressure of the plant cell. To adapt to such stresses, plant cells produce stabilizing osmolytes to increase their salt tolerance (*Hanson et al., 1994*). Several MAPK factors have been reported in *A. thaliana*, including ATMAPKK1, which is upregulated under Cd stress (*Suzuki, Koizumi & Sano, 2001*). The accumulation of ROS activates the MPK3 and MPK6 response under Cd stress (*Liu et al., 2010*). In *Oryza sativa*, MAPK cascades have also been reported in response to Cd stress, including *OsMKK4, OsMSRMK2, OsMSRMK3, OsWJUMK, OsMPK2, OsMPK3*, and *OsMPK6* (*Agrawal, Rakwal & Iwahashi, 2002*; *Agrawal et al., 2003*; *Yeh, Hsiao & Huang, 2004*; *Yeh,*
*Chien & Huang, 2007*). Moreover, SAMK, SIMK, MMK2, and MMK3 were upregulated under Cu/Cd stress in *M. sativa* (*Jonak, Nakagami & Hirt, 2004*). ZmMPK3 exhibited upregulation based on mRNA levels in *Z. mays* (*Wang et al., 2010*). In our study, MAPKKK3 and MAPKKK12 were enhanced in creeping bentgrass in response to Cd stress. MAPKKK12 was upregulated in the BT2_5 and BT43 treatments (Table S6). MAPKKK12 may play an important role in activating tolerance to Cd stress in creeping bentgrass.

## WRKY TFs play important roles in the creeping bentgrass response to Cd stress

WRKY family members contain highly conserved domains, which play many roles in the regulation of gene expression, including under drought, salinity, and nutrient starvation conditions (*Chen et al., 2012*). Several WRKY TFs are important components of plant signal transduction during abiotic stress responses. Overexpression of *OsWRKY08* in *A. thaliana* resulted in increased tolerance to mannitol stress through increased root production in terms of number and length (*Song, Jing & Yu, 2009*). Overexpression of *OsWRKY72* in *A. thaliana* showed that it is involved in multiple physiological processes and interferes in the ABA signal and auxin transport pathway (*Song et al., 2010*). Three *GmWRKY* genes from soybean were investigated in *A. thaliana*, and overexpression of *GmWRKY21* increased tolerance to cold stress, *GmWRKY54* was involved in salt and drought tolerance, and *GmWRKY13* increased tolerance to salt stress (*Zhou et al., 2008*). *WRKY75* was determined to regulate the nutrient starvation response in *A. thaliana*. Our study showed that six WRKY TFs are involved in Cd stress in creeping bentgrass. *WRKY23* and *WRKY75* were quickly upregulated from lower to higher Cd concentrations (Table S9), indicating that *WRKY23* and *WRKY75* could increase tolerance to Cd stress. However, *WRKY12*, *WRKY2*, and *WRKY53* were downregulated from lower to higher Cd concentrations (Table S9), suggesting negative regulatory roles in the response to Cd stress.

## Role of bZIP TFs in the creeping bentgrass response to Cd stress

The bZIP TFs make up a large family, of which many genes are expressed in plants under abiotic stress (*Hurst, 1994*; *Vetten & Ferl, 1995*). In *A. thaliana*, the TFs *AtbZIP1–AtbZIP75* were reported to regulate a variety of biological processes such as stress signaling, light response, and seed development. Among these TFs, seven have been identified as being involved in stress signaling: *AtbZIP39*, *AtbZIP36*, *AtbZIP38*, *AtbZIP66*, *AtbZIP40*, *AtbZIP35*, and *AtbZIP37* (*Jakoby et al., 2002*; *Choi et al., 2000*; *Finkelstein & Lynch, 2000*; *Lopez-Molina, Mongrand & Chua, 2001*; *Uno et al., 2000*). Overexpression of the *SlAREB1* gene in the bZIP family led to upregulation under salt stress in tobacco (*Yánez et al., 2009*). The *ZmbZIP72* gene from maize was overexpressed in *A. thaliana*, which showed that *ZmbZIP72* functions as an ABA-dependent TF and increases abiotic stress tolerance (*Ying et al., 2012*). In the present study, three bZIP family genes were obtained from creeping bentgrass under Cd stress, including *bZIP06*, *bZIP43*, and *bZIP19* (Table S10). *bZIP43* was upregulated throughout the Cd treatment process, while *bZIP06* and *bZIP19* genes were upregulated at the lower concentration (BT2_5) and downregulated at the higher concentration (BT43). In summary, *bZIP43* may play a positive regulatory role under Cd stress.

## ERF TFs expressed in response to Cd stress

The expression of *ERF* genes plays crucial roles in plant stress responses. For example, *AtERF73/HRE1* in *A. thaliana* plays a negative regulatory role in response to ethylene (*Yang et al., 2011*). Overexpression of *TSRF1* in the ERF family enhances osmotic and drought tolerances in rice (*Quan et al., 2010*; *Oono et al., 2014*). TaERF3 activates several genes related to stress that enhance the ability of wheat to adapt to salt and drought stresses (*Rong et al., 2014*). Overexpression of *OsERF922* decreased tolerance to salt stress in *M. oryzae* (*Liu et al., 2012*). Meanwhile, overexpression of *GmERF3* increased salt and drought tolerance in tobacco (*Zhang et al., 2009*). Moreover, overexpression of *NtERF5* enhanced resistance to tobacco mosaic virus (*Fischer & Dröge-Laser, 2004*). In summary, ERF TFs and *ERF* genes play important roles in biological and abiotic stress responses. Our RNA-Seq data showed that seven DEGs involved in Cd stress in creeping bentgrass, *ERF1B*, *ERF110*, *ERF7*, *ERF113*, and *ERF15*, were upregulated from lower to higher Cd concentrations (Table S11). These ERF genes may play positive regulatory roles in the response to Cd stress in creeping bentgrass. On the other hand, the *ERF4* gene was downregulated in all Cd treatments, and may play a negative role in Cd stress.

## Roles of MYB TFs in Cd stress

MYB proteins have a highly conserved DNA-binding domain called the MYB domain (*Martin & Paz-Ares, 1997*) and have been widely investigated in plant species such as *A. thaliana*, rice (*O. sativa*), and maize (*Chen et al., 2006*; *Oono et al., 2016*), (*Yue et al., 2016*). Several MYB TFs are involved in abiotic stresses; for example, overexpression of *DwMYB2* in *A. thaliana* suppressed iron transport from root to shoot (*Chen et al., 2006*). The function of *MxMYB1* has been identified as negative regulation of iron uptake and storage in *A. thaliana* (*Shen et al., 2008*). Overexpression of *MYB15* enhanced drought and salt tolerance in *A. thaliana* (*Ding et al., 2009*). Overexpression of *MdoMYB121* enhanced tolerances to salinity and drought in tomato and apple plants (*Cao et al., 2013*). Overexpression of *TaMYB33* and *TaMYB73* enhanced salt and drought tolerances (*Qin et al., 2012*; *He et al., 2011*). Moreover, overexpression of *OsMYB48-1* in rice promoted drought and salinity tolerance through regulation of ABA synthesis (*Xiong et al., 2014*). In our study, four DEGs in the MYB family were enhanced under Cd stress (*MYB39*, *MYB108*, *MYB305*, and *MYB4*). All MYB TF genes were upregulated more than two-fold in the higher Cd treatment (BT43). Moreover, in BT2_5, *MYB39*, *MYB108*, and *MYB305* were upregulated, while *MYB4* was downregulated (Table S12). Our results indicated that *MYB39*, *MYB108*, and *MYB305* might play positive roles in the response of creeping bentgrass to Cd stress.

## CONCLUSIONS

This study investigated the molecular characteristics of DEGs under Cd stress based on transcriptomic analysis. A total of 463,184 unigenes were obtained from creeping bentgrass leaves. Changes in leaf tissue morphology were observed, revealing that higher Cd concentrations damage leaf tissues. Moreover, four key TF families (WRKY, bZIP, ERF, and MYB) were involved in Cd stress in creeping bentgrass. These TFs exhibited active expression in RNA-Seq data. For example, ERF115 was upregulated more than five-fold.

Previous research has shown that these TFs are involved in abiotic stress responses. We found that these four TFs may play crucial roles in the response of creeping bentgrass to Cd stress. This study provides a novel perspective for elucidating the molecular mechanisms of the response of creeping bentgrass to Cd stress, and provides important reference information for phytoremediation.

## ACKNOWLEDGEMENTS

We would like to thank Vladimir Uversky academic editor and two anonymous reviewers for their comments that helped to greatly improve the manuscript.

### Funding

This work was supported by the Scientific Technology Plan Program of Shenzhen (No. JCYJ20160331151245672). The funders had no role in study design, data collection and analysis, decision to publish, or preparation of the manuscript.

### Grant Disclosures

The following grant information was disclosed by the authors:
Scientific Technology Plan Program of Shenzhen: JCYJ20160331151245672.

### Competing Interests

The authors declare there are no competing interests.

### Author Contributions

- Jianbo Yuan conceived and designed the experiments, performed the experiments, analyzed the data, prepared figures and/or tables, authored or reviewed drafts of the paper, approved the final draft.
- Yuqing Bai approved the final draft.
- Yuehui Chao performed the experiments, analyzed the data, prepared figures and/or tables, approved the final draft.
- Xinbo Sun contributed reagents/materials/analysis tools, authored or reviewed drafts of the paper, approved the final draft.
- Chunyan He contributed reagents/materials/analysis tools, approved the final draft.
- Xiaohong Liang prepared figures and/or tables, approved the final draft.
- Lijuan Xie conceived and designed the experiments, performed the experiments, analyzed the data, prepared figures and/or tables, approved the final draft.
- Liebao Han analyzed the data, approved the final draft.

### Data Availability

NCBI accession number SRP153347.

## Supplemental Information

Supplemental information for this article can be found online at http://dx.doi.org/10.7717/peerj.5191#supplemental-information.

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
