# Peer review of "Genome-wide analysis reveals four key transcription factors associated with cadmium stress in creeping bentgrass (Agrostis stolonifera L.)"

_PeerJ, doi:10.7717/peerj.5191_

## Round 0.1 · original submission · Major Revisions

As you can see, both reviewers have found your manuscript interesting. However, several shortcomings were indicated too. Please carefully address critical issues raised by both reviewers. Pay special attention to the comments of reviewer #2, who gave several useful suggestions.

Reviewer 1 ·

Basic reporting

The manuscript provides a thorough and motivating view of existing literature to support the importance and relevance of the present study. The language is clear, and all but figures 1-3 are self-contained and provide sufficient information. Figures 1-3 could use more information in caption to describe them adequately.

Experimental design

Overall, the discussion and results of the studies is well written, very clear, and provides sufficient information. Materials and methods are also described with sufficient information.

Validity of the findings

Impact, novelty, and description of results is well stated, and relevance to answering the central research question of the manuscript is clearly determined.

Additional comments

The manuscript by Yuan et al. studies and analyzes the role played by four transcription factor families associated with Cadmium stress in creeping bentgrass. The manuscript provides a thorough and motivating view of existing literature to support the importance and relevance of the present study. The discussion and results of the studies is well written, very clear, and provides sufficient information. The manuscript can be accepted for publication after addressing the following minor comments:
1. The authors can mention briefly in the text their rationale for using KEGG database for analysis
2. Figures 1-3 need more information in caption to describe them adequately

Reviewer 2 ·

Basic reporting

It is highly recommended that the authors rewrite the manuscript, ideally, with some help from a native English speaker. English may not be the first language for the authors, but in the current form, it is very difficult to understand the points authors are trying to convey. Sentences are incomplete and grammatically incorrect. There is excessive use of colloquial language instead of scientific terms – which makes it hard to interpret the context and research findings.

Examples:
Abstract:
“Studying the uptake, translocation, and accumulation of Cd by plants is crucial for cultivation of pollution-tolerant plant species.”

How the findings in this study could be applied towards cultivation of pollution-tolerant plant species is not discussed anywhere. Without any conclusive results, making such overreaching claims is hasty.


Repetitive sentences:
Abstract:
“The main objective of this study was to reveal differentially expressed genes (DEGs) under lower (BT2_5) and higher (BT43) Cd
concentration treatments in creeping bentgrass. This study includes characterization of DEGs from creeping bentgrass leaves under Cd stress.”

Ambiguous pronouns:
Abstract:
“This study is the first report of the molecular characteristics of DEGs under Cd stress using transcriptomic analysis in
creeping bentgrass. These results provide new insight into the key roles played by these TFs under different Cd stress conditions and provide an important reference for breeders cultivating pollution-tolerant species.”

Which ‘these’ results are referred to here?


Sentence structures are very confusing:
Line 33-34
“As an increasing factor in environmental pollution, cadmium (Cd) pollution has once again attracted public attention.”


Write shorter sentences (Line 34-36):
“Cd is a non-essential element that is highly toxic and easily accumulates in plants, inhibiting plant development through its effects on physiological and metabolic processes (Nawrot et al., 2006; Hasan et al., 2009; DalCorso et al., 2010).”

Break up such sentence into 2-3 shorter sentences. Short and clear sentences will help convey point easily.


Unclear study objectives:
“Studying the uptake, translocation, and accumulation of Cd by plants is crucial for cultivation of pollution-tolerant plant species.”

What is the scope of the current study?

Manuscript structure:
Several references are missing.
Several studies have investigated differentially expressed genes in response to Cd stress in different plant species.
Yue R, Lu C, Qi J, et al. Transcriptome Analysis of Cadmium-Treated Roots in Maize (Zea mays L.). Frontiers in Plant Science. 2016;7:1298. doi:10.3389/fpls.2016.01298.
Oono Y, Yazawa T, Kanamori H, et al. Genome-Wide Transcriptome Analysis of Cadmium Stress in Rice. BioMed Research International. 2016;2016:9739505. doi:10.1155/2016/9739505.
Oono Y, Yazawa T, Kawahara Y, et al. Genome-Wide Transcriptome Analysis Reveals that Cadmium Stress Signaling Controls the Expression of Genes in Drought Stress Signal Pathways in Rice. Tran L-SP, ed. PLoS ONE. 2014;9(5):e96946. doi:10.1371/journal.pone.0096946.

Figures:
Please rewrite all the figure legends. (Once published) readers may (and most likely) try to understand the research primarily by looking at figures. Do the figure legends in their current format help readers interpret your results?

Please include in the introduction section, a brief summary of all the studies you discuss in the discussion section.
Results section needs to be more structured. In its current format, the results section reads like a materials and methods section.
Discussion reads like a results section.

Please look at the following article for manuscript structure and wiring:
Scientific reports: Transcriptome analysis of molecular mechanisms responsible for light-stress response in Mythimna separata (Walker)

Experimental design

No rationale for study design has been provided:
Why did you select 7-day treatment?
Why did you select 2.5 mM and 43 mM Cd concentrations? Concentrations in mM ranges are very high – are these environmentally relevant?
See the following studies – Cd concentrations in µM ranges:
Song J, Feng SJ, Chen J, Zhao WT, Yang ZM. A cadmium stress-responsive gene AtFC1 confers plant tolerance to cadmium toxicity. BMC Plant Biology. 2017;17:187. doi:10.1186/s12870-017-1141-0.
Oono Y, Yazawa T, Kawahara Y, et al. Genome-Wide Transcriptome Analysis Reveals that Cadmium Stress Signaling Controls the Expression of Genes in Drought Stress Signal Pathways in Rice. Tran L-SP, ed. PLoS ONE. 2014;9(5):e96946. doi:10.1371/journal.pone.0096946.

“Hoagland nutrient solution was used for irrigation”. Either cite a reference or include details.

After 3 months, mature plants were obtained and subjected to 0 mM (CK), 2.5 mM (BT2_5), and 43 mM (BT43) CdCl2 treatments.
How exactly was this ‘subjecting’ done? Provide details.

Validity of the findings

Methods section indicates that “After 3 months, mature plants were obtained and subjected to 0 mM (CK), 2.5 mM 88 (BT2_5), and 43 mM (BT43) CdCl2 treatments.” Whereas the results section indicates the effect of Cd stress on ‘development’. To study the effect on development, samples need to be collected at different time points after exposure to Cd.

Line 357:
No justification of how upregulation of certain genes is related to tolerance is provided.
MAPKKK12 may play an important role in activating tolerance to Cd stress in creeping bentgrass.
Line 372-373:
WRKY23 and WRKY75 were quickly upregulated from lower to higher Cd concentrations (Table S9), indicating that WRKY23 and WRKY75 could increase tolerance to Cd stress.

This study is the first report of the molecular characteristics of DEGs under Cd stress based on a transcriptomic analysis.
This is a false claim. Check the publications that you have not cited. Maybe in Agrostis stolonifera L, but that is not what the sentence states.

---

## Round 0.2 · accepted · Accept

In my view, you did a very good job addressing the reviewers' comments and revising your manuscript to fix all the issues indicated by reviewers. Thank you very much for all your efforts to improve this manuscript.

#